# Hospital bed capacity and usage across secondary healthcare providers in England during the first wave of the COVID-19 pandemic: a descriptive analysis

Bilal Akhter Mateen [1,2,3] Harrison Wilde [4] John M Dennis [5]
Andrew Duncan,[2,6] Nick Thomas,[5,7] Andrew McGovern,[5,7] Spiros Denaxas [2,3,8]
Matt Keeling,[9] Sebastian Vollmer[2,4]

► Prepublication history and additional materials for this paper are available online. To view these files, please visit the journal online (http://dx.doi. org/10.1136/bmjopen-2020- 042945).

For numbered affiliations see end of article.

**Correspondence to**
Dr Bilal Akhter Mateen;
bilal.mateen@nhs.net

## ABSTRACT

**Objective** In this study, we describe the pattern of bed occupancy across England during the peak of the first wave of the COVID-19 pandemic.

**Design** Descriptive survey.

**Setting** All non-specialist secondary care providers in England from 27 March27to 5 June 2020.

**Participants** Acute (non-specialist) trusts with a type 1 (ie, 24 hours/day, consultant-led) accident and emergency department (n=125), Nightingale (field) hospitals (n=7) and independent sector secondary care providers (n=195).

**Main outcome measures** Two thresholds for 'safe occupancy' were used: 85% as per the Royal College of Emergency Medicine and 92% as per NHS Improvement.

**Results** At peak availability, there were 2711 additional beds compatible with mechanical ventilation across England, reflecting a 53% increase in capacity, and occupancy never exceeded 62%. A consequence of the repurposing of beds meant that at the trough there were 8.7% (8508) fewer general and acute beds across England, but occupancy never exceeded 72%. The closest to full occupancy of general and acute bed (surge) capacity that any trust in England reached was 99.8% . For beds compatible with mechanical ventilation there were 326 trust-days (3.7%) spent above 85% of surge capacity and 154 trust-days (1.8%) spent above 92%. 23 trusts spent a cumulative 81 days at 100% saturation of their surge ventilator bed capacity (median number of days per trust=1, range: 1–17). However, only three sustainability and transformation partnerships (aggregates of geographically co-located trusts) reached 100% saturation of their mechanical ventilation beds.

**Conclusions** Throughout the first wave of the pandemic, an adequate supply of all bed types existed at a national level. However, due to an unequal distribution of bed utilisation, many trusts spent a significant period operating above 'safe-occupancy' thresholds despite substantial capacity in geographically co-located trusts, a key operational issue to address in preparing for future waves.

## INTRODUCTION

The ability of hospitals to cope with large influxes of patients, either due to a pandemic illness or seasonal increases in respiratory

## Strengths and limitations of this study

► The use of an administrative data that are collected by the statutory regulator as part of its legal mandate resulted in minimal missing information.

► Results are presented in the context of several geographical units of healthcare provision (ie, hospital/ site, trust, and sustainability and transformation partnership level), thus providing a much richer understanding of resource utilisation that is less prone to the diluent effects of higher level geographies.

► The data represent a daily snapshot and therefore are unable to capture the nuances of the hospital throughput; in essence, both under-reporting and over-reporting of occupancy are possible using this method.

► The use of the occupancy thresholds reflects a limitation of our analysis, in that a proxy for adverse outcomes had to be used given that the necessary information was not readily available to directly explore the relationship between occupancy and patient-level outcome.

► The results of this study may not be generalisable to other countries given that it is specific to the UK National Health System infrastructure.

disease exacerbations, is in part dictated by the availability of beds.[1] Since 1987, when formal reporting of the number of hospital beds began in the UK, there has been a sustained decline in the number of available beds across the National Health Service (NHS).[2] In recent years, this issue has garnered more attention due to the annual 'winter bed crisis',[3 4] where the end of the calendar year heralds a surge in emergency admissions often resulting in hospitals operating well above quality and operational performance tipping points, that is, 85% or 92% total bed occupancy.[5–7] The saturation of hospital beds is not only problematic through its impact on the ability of the workforce to

deliver high-quality care,[8] but additionally the bottle-necking of the emergency care workflow has been shown to contribute to suboptimal outcomes for patients,[9] including increased numbers of healthcare-acquired infections[10] and increased mortality.[11–13]

These concerns about the NHS' ability to cope with large influxes of patients took on a new level of significance in early 2020, when the WHO formally declared COVID-19 a pandemic illness, due to its virulence and the magnitude of the disease's impact globally.[14] As early reports from China were published, it became apparent that a relatively large proportion of individuals who contracted COVID-19 required admission to hospital,[15] for example due to new oxygen requirements, sepsis, acute respiratory distress syndrome and even multiorgan dysfunction. Forecasts of the potential number of people requiring hospital admission and mechanical ventilation across the UK suggested that the baseline capacity of the NHS would be insufficient.[16] In an effort to ensure sufficient capacity the British government instituted a series of policies, including facilitating the discharge of individuals who had been delayed due to non-medical reasons in an effort to unlock capacity,[17] cancelling all non-urgent clinical work, opening large field hospitals (ie, the Nightingale hospitals)[18] and increasing mechanical ventilator availability for use in clinical areas repurposed to manage patients requiring higher-acuity care.[19]

The UK started making significant strides towards rolling back its non-pharmacological interventions in June 2020, including reopening schools and planning for the discontinuation of shielding for vulnerable people,[20] signalling an end to the first wave of the pandemic.[21] Following these changes, there was the potential for a second wave of COVID-19 related admissions at the end of 2020. Understanding regional differences in hospital capacity is fundamental to informing the UK's response to the potential second wave andany future epidemics, as well as for elucidating how to safely wind down repurposed surge capacity, such as operating theatres to allow other much needed clinical activity to restart.[22] However, other than a few isolated news reports of hospitals exceeding their ventilator capacity,[23] it is unclear how well the NHS as a whole managed to respond to the additional demand for beds over the recent months. In this study, we sought to describe the pattern of bed occupancy in hospitals across England during the first wave of the COVID-19 pandemic.

## METHODS
### Primary data source
Data were accessed from the daily situation reports ('SitReps', covering the previous 24 hours) provided to the Scientific Pandemic Influenza Group on Modelling by NHS England on behalf of all secondary care providers. All NHS acute care providers, independent sector care providers and field hospitals in England submitting information to the daily situation reports were eligible for inclusion.

### Study population
The data are presented in the context of several different units of secondary care provision: hospitals/sites, trusts, sustainability and transformation partnerships (STPs; aggregates of geographically co-located trusts), regions and the whole of England (ie, national), where each is an aggregate of the preceding unit (the structure of UK care providers is explained in the online supplemental material).

### Inclusion and exclusion criteria
Exclusions were applied at the trust level for NHS-specific care providers. Exclusion criteria were as follows: acute specialist trusts: women and/or children (n=4), neurology and ophthalmology (n=2), heart and lung (n=3), orthopaedic, burns and plastics (n=4), and cancer (n=3). The remaining care providers were grouped into three categories and analysed separately: (1) acute (non-specialist) trusts with a type 1 (ie, 24 hours/day, consultant-led) accident and emergency department (n=125); (2) Nightingale (field) hospitals (n=7); and (3) independent sector providers (n=195).

### Recruitment period
Data were available from 27 March 2020 (the first available SitRep) to 5 June 2020 inclusive.

### Recorded information
The data specification comprised resource utilisation and capacity-specific information, including the number of beds at each trust, stratified by several factors of interest, including acuity and COVID-19 ascertainment (further defined in online supplemental material). Notably, beds were only recorded as being available if they were 'funded' (ie, there was adequate staffing and resources for the bed to be occupied), so as to prevent counting of beds that could not accommodate a new patient. Bed acuity was organised into general and acute (G&A), beds compatible with non-mechanical ventilation and beds compatible with mechanical ventilation. Occupancy is calculated based on the status of each bed at 08:00 each day, and then later separated by the proportion that had a positive COVID-19 test; there was no available information on the temporal relationship between admission and a positive test and thus these data reflect some combination of community-acquired and nosocomial COVID-19.

Reporting fields changed on 27 April 2020, with several additional columns being added, which included specific fields for level 2 (HDU: High Dependency Unit) and level 3 (ICU: Intensive Care Unit) beds. The impact of these changes is detailed in the online supplemental material. However, one crucial outcome was that it became apparent the definition of critical care beds used prior to 27 April 2020 was not consistent with prior reporting practices of only including level 2 (HDU) and level 3 (ICU) beds,[24] as the newly reported values did not equal

the simultaneously reported critical care values. As such, any results pertaining to critical care, HDU and ICU are reported separately in the online supplemental material.

NHS England reports trust-level data, whereas we additionally attempted to disaggregate this information into the individual hospitals that the trusts comprise. Not all of the trusts were amenable to disaggregation from the trust-level data into independently reported sites in the available extracts, resulting in a final sample of 173 unique hospital sites, comprising 91.7% of the total number of ventilated beds and 81.4% of the G&A beds when compared with trust level. The change in data reporting introduced on 27 April 2020 also resulted in variation in information capture; for data prior to 27 April, the results available reflect 89.6% of all mechanical ventilator beds and 86.9% of G&A beds, whereas for data from 27 April onwards the results reflect 93.0% of all mechanical ventilator beds but 77.0% of G&A beds.

### Outcome

The primary outcomes of interest were bed availability and bed occupancy by patients with and without COVID-19, for each level of secondary care provision, that is, hospital, trust and STP (aggregates of geographically co-located trusts). Different 'safe occupancy' thresholds were used to interpret the results: 85% as per the Royal College of Emergency Medicine and 92% as per NHS Improvement. We also compared occupancy against baseline bed occupancy (see online supplemental material for definitions) and 100% of surge capacity.

### Statistical analysis

We generated and reported descriptive summaries (eg, medians, ranges, counts, proportions) of the data. We reported absolute numbers for hospital, trusts and STPs attaining specific occupancy thresholds. In light of the discordant critical care and HDU/ICU values, this analysis was handled and reported separately (see online supplemental material). To capture the temporal aspect of the information available, the number of hospital-days, trust-days and STP-days spent above hospital baseline capacity and surge capacities of 85%, 92% and 100% is also reported. Full details on the quality control procedures are reported in online supplemental SFigures 1 and 2. Details on aggregation and disaggregation of geographical information are provided in online supplemental STable 1 and 2. Analyses were carried out in R,[25] and figures were generated using the ggplot2 package.[26] Maps were acquired from the UK's Office for National Statistics Open Geography Portal.[27]

### Patient and public involvement

No patients were involved in the design of the study, interpretation of the results or drafting of this manuscript.

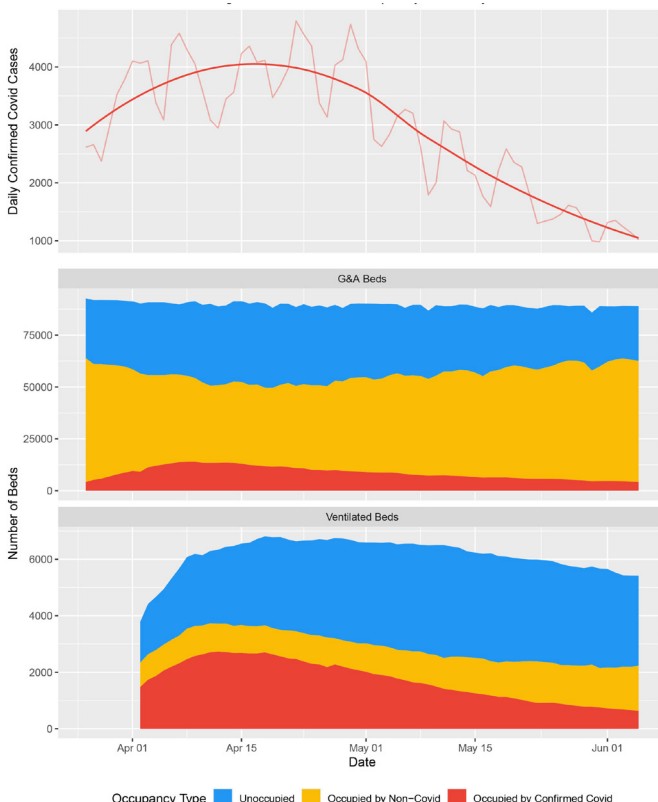

**Figure 1** National and regional bed occupancy. (Top) An epidemic curve showing the number of confirmed cases of COVID-19 across England based on the date that the specimen was taken; raw data are available at https://coronavirus.data.gov.uk/details/cases?areaType=nation&areaName=England. The superimposed highly saturated solid line represents a smoothened function of the raw data, whereas the less saturated solid line represents the underlying raw values. The former is based on the ggplot loess fit for trend lines, using local polyregression curve fitting. (Middle) Total capacity and occupancy status for general and acute (G&A) beds at the national level over the course of the first wave. (Bottom) Total capacity and occupancy status for beds compatible with mechanical ventilation at the national level.

### RESULTS

#### National mobilisation

During the first wave of the pandemic, the NHS repurposed general/acute beds into those suitable for higher acuity patients (ie, HDU/ICU) and patients requiring mechanical ventilation. Available ventilated bed capacity peaked at an additional 2711 beds, a 53% increase from a baseline of 4123 beds. Ventilated beds occupancy never exceeded 62% of this capacity at a national level (figure 1), and the proportion of occupied beds which contained patients with COVID-19 fluctuated between 30.4% and 76.0% over the course of the first wave; however, there were notable regional differences in COVID-19-specific demand (figure 2, online supplemental SFigure 3). Similar patterns were observed in critical care/HDU and ICU beds (online supplemental SFigure 4). A consequence of the repurposing of beds

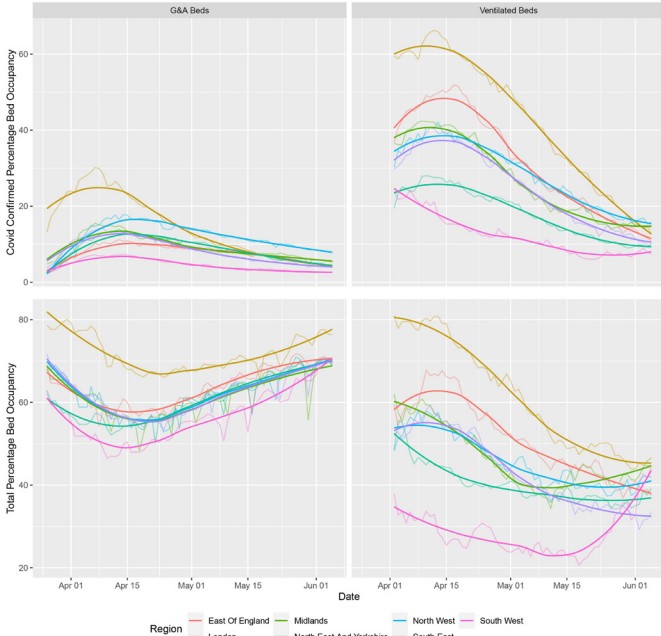

**Figure 2** Total bed occupancy in each of the seven regions of England. (Top) COVID-19-specific occupancy in each of the seven regions across England for both general and acute (G&A; left) beds and beds compatible with mechanical ventilation (right). (Bottom) Total occupancy (COVID-19 positive and negative) in each of the seven regions across England for both G&A (left) beds and beds compatible with mechanical ventilation (right). The highly saturated solid line represents a smoothened function of the raw data, whereas the less saturated solid line represents the underlying raw values. The former is based on the ggplot loess fit for trend lines, using local polyregression curve fitting.

for higher acuity patients, there was 8.7% reduction (n=8508) in G&A beds from a baseline of 97 293 available beds. There was a large reduction in the number of beds occupied by patients without COVID-19; 53 136 fewer beds were occupied (58.8% reduction) at the nidus

compared with the average occupancy from January to March 2020. Total bed occupancy never exceeded 72% nationally (figure 1). Data were relatively complete over the observation period (from 27 March to 5 June 2020), with no unavailable records for COVID-19-specific occupancy across G&A and mechanical ventilation compatible beds and less than 10% for non-COVID-19/unoccupied beds (see online supplemental material).

### Occupancy relative to baseline capacity

Out of the 125 trusts (aggregates of hospitals), 3 trusts (2.4%) at some point during the first wave were operating above their baseline bed availability for G&A beds (124 trust-days (1.4% of the total 8738 days at risk); median number of days per trust=36 days (range: 30–58); online supplemental SFigure 5). For beds compatible with mechanical ventilation, 87 trusts (69.6%) at some point during the first wave were operating above their baseline bed availability (2456 trust-days (28.1% of the total at risk); median number of days per trust=24 days (range: 1–61); online supplemental SFigure 6). Similar results to that of mechanical ventilation compatible beds were seen for critical care/HDU and ICU bed occupancy (see online supplemental material, SFigure 7 and 8).

### Occupancy relative to surge capacity

Table 1 summarises the number of hospitals, trusts and STPs operating above the prespecified thresholds for 'safe occupancy' and details the duration (ie, median number of days) that each spent above the designated thresholds.

### Hospital-level occupancy

Of the total 11 851 English hospital-days at risk over the study period, 494 hospital-days (4.17% of the total days at risk) were at or above 85% of G&A bed (surge) capacity, 110 hospital-days (0.92%) were at or above 92% of G&A bed (surge) capacity, and only 10 were spent at 100% of G&A surge capacity (figure 3). Similarly, for beds

**Table 1** Number of hospital/trusts/STPs at each occupancy threshold for different bed types

| Bed type | Organisational unit | Occupancy threshold | | | | | |
|---|---|---|---|---|---|---|---|
| | | **>85%** | | **>92%** | | **100%** | |
| | | n (%) | Median number of days at or above threshold (range) | n (%) | Median number of days at or above threshold (range) | n (%) | Median number of days at or above threshold (range) |
| General and acute | Hospital/site (n=173) | 56 (32.4) | 6 (1–45) | 19 (11.0) | 3 (1–19) | 1 (0.6) | 10 |
| | Trust (n=125) | 30 (24.0) | 5 (1–46) | 14 (11.2) | 3 (1–13) | 0 (0.0) | – |
| | STP (n=42) | 2 (4.8) | 10 (3–17) | 2 (4.8) | 1 (1–1) | 0 (0.0) | – |
| Mechanical ventilation | Hospital/site (n=173) | 91 (52.6) | 4 (1–48) | 72 (41.6) | 3 (1–48) | 52 (30.0) | 2 (1–48) |
| | Trust (n=125) | 58 (46.4) | 3 (1–27) | 40 (32.0) | 2 (1–17) | 23 (18.4) | 1 (1–17) |
| | STP (n=42) | 10 (23.8) | 2 (1–11) | 5 (10.4) | 1 (1–6) | 3 (7.1) | 1 (1–2) |

STP, sustainability and transformation partnership.

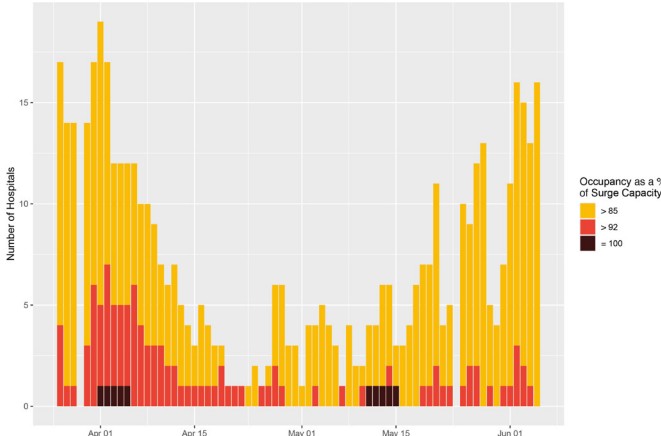

**Figure 3** Hospital-level general and acute bed occupancy (based on surge capacities) across England. The number of hospitals with general and acute bed occupancy in excess of the thresholds for 'safe and effective' functioning, that is, 85% as defined by the Royal College of Emergency Medicine,[6] and 92% as defined by NHS Improvement and NHS England (yellow and red, respectively),[7] across England, from 26 March to 5 June. All data were missing for 29 March and 24 May. NHS, National Health Service.

compatible with mechanical ventilation there were 586 hospital-days (4.94%) spent above 85% of surge capacity, 320 hospital-days (2.70%) were spent above 92%, and 226 hospital-days (1.9%) were spent at 100% occupancy (see figure 4). Summaries of the size and geographical locations of hospitals stratified by saturation are in online supplemental STable 3.

### Trust-level bed occupancy

Over the study period, there were 287 trust-days (3.3% of the total days at risk) where G&A bed occupancy

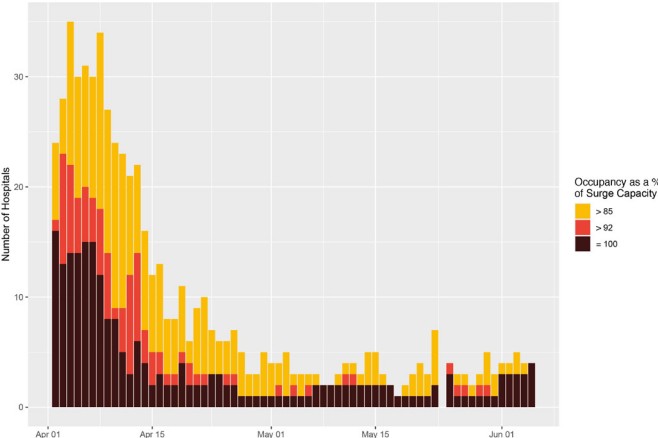

**Figure 4** Hospital-level ventilator bed occupancy (based on baseline capacities) across England. The number of hospitals with occupancy of mechanical ventilation beds in excess of the thresholds for 'safe and effective' functioning, that is, 85% as defined by the Royal College of Emergency Medicine,[6] and 92% as defined by NHS Improvement and NHS England (yellow and red, respectively),[7] across England, from 1 April to 5 June. All data were missing for 24 May. NHS, National Health Service.

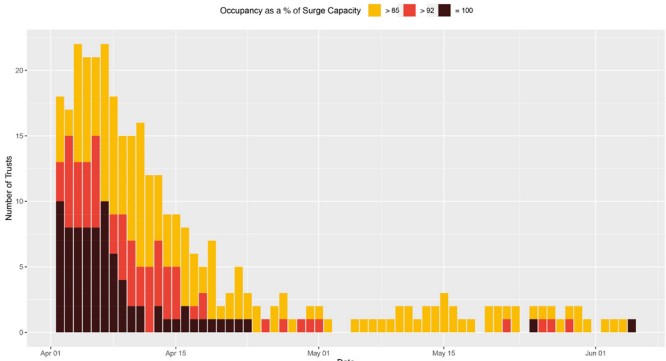

**Figure 5** Trust-level ventilator bed occupancy (based on surge capacities) across England. The number of trusts with occupancy of mechanical ventilation beds in excess of the thresholds for 'safe and effective' functioning, that is, 85% as defined by the Royal College of Emergency Medicine,[6] and 92% as defined by NHS Improvement and NHS England (yellow and red, respectively),[7] across England, from 26 March to 5 June. All data were missing for 29 March and 24 May. Several hospitals reported values consistent with 100% occupancy (black). NHS, National Health Service.

exceeded 85% of surge capacity and 57 trust-days (0.7%) were at or above 92% of bed (surge) capacity. The closest to capacity any trust in England reached was 99.8% for G&A beds. However, for beds compatible with mechanical ventilation there were 326 trust-days (3.7%) spent above 85% of surge capacity and 154 trust-days (1.8%) spent above 92%. There were 23 trusts that reached 100% saturation of their mechanical ventilator bed capacity (figure 5, online supplemental SFigure 9).

### STP-level bed occupancy

Across the 42 STPs (aggregates of geographically co-located trusts), there were 20 STP-days (0.7% of the total days at risk) where G&A bed occupancy exceeded 85% of surge capacity. The highest any STP reached for G&A bed occupancy was 92.7%. For beds compatible with mechanical ventilation, there were 35 STP-days (1.2%) where occupancy exceeded 85% of surge capacity, 11 STP-days (0.4%) in excess of 92% occupancy and 4 STP-days (0.1%) at full occupancy (all of which were for STPs outside London: (1) Somerset, (2) Suffolk and North East Essex, and (3) Shropshire, Telford and Wrekin; online supplemental SFigure 10). Figure 6 illustrates the number of STPs operating at each distinct occupancy threshold as a proportion of baseline and actual surge capacity. The full time-lapse for G&A (online supplemental video 1) and ventilator bed (online supplemental video 2) occupancy over the period of interest can be found in the online supplemental material. A similar pattern was seen in the context of critical care/HDU and ICU beds across the STPs (online supplemental SFigure 7 and 8).

### Field (Nightingale) hospital occupancy

Of the reported bed capacity achievable through opening the Nightingale hospitals, at peak occupancy only 1.23% of the theoretical maximum were

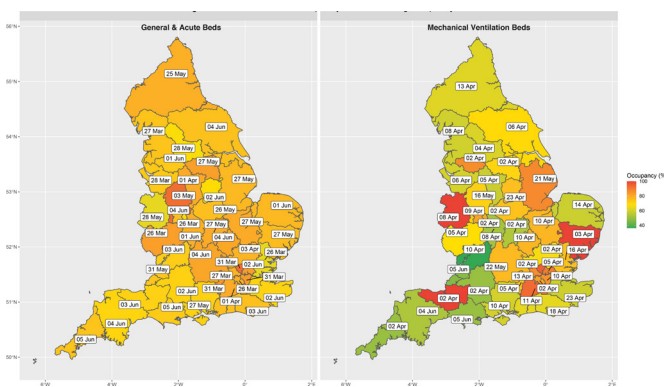

**Figure 6** Peak STP bed occupancy across England. The date on which general and acute bed occupancy (left) and mechanical ventilator beds (right) peaked, based on surge capacities at the STP level across England. The geotemporal pattern of peak occupancy clearly demonstrates that there was always residual general and acute bed capacity at the STP level and that all regions across England experienced similar levels of saturation. However, saturation of mechanical ventilator beds differed substantially by location. STP, sustainability and transformation partnership.

being used (table 2). This equates to 618 bed days for patients with COVID-19 requiring mechanical ventilation and 1483 bed days for all other types of intervention for patients with COVID-19 (ie, oxygenation, non-invasive respiratory support, non-respiratory organ support and so on).

### Independent sector care providers

Variations in reporting meant that the number of providers reporting each day varied, with a median of 181 providers (range: 172–187). At peak occupancy, no more than 134 independent sector beds were occupied with patients who were confirmed COVID-19-positive. With regard to patients without COVID-19, at peak occupancy there were 1350 people in independent sector beds, representing a peak saturation of 18.7% (based on the total number of beds reported during contractual negotiations). In summary, there

were 3360 bed days for patients with confirmed COVID-19 accommodated by the independent sector (86 mechanical ventilator bed days, 104 non-invasive ventilation bed days and 3170 other bed days) and 53 937 bed days for patients without COVID-19 (2771 mechanical ventilator bed days, 2046 non-invasive ventilation bed days and 49 120 other bed days) between 2 April and 5 June across England.

### DISCUSSION

This national study of hospital-level bed occupancy provides unique insight into the impact of COVID-19 on bed-specific resource utilisation across an entire country. Our analysis suggests that the response of the NHS and British government to COVID-19 was sufficient to alleviate early concerns regarding the number of mechanical ventilators and critical care beds at a national level; however, local variation in demand (ie, regional variation in COVID-19 prevalence) still meant that many trusts reached 100% capacity for both. Moreover, examining occupancy in the context of different organisational units (ie, trust level vs STP level) suggests that the higher order networks (ie, STPs) were not efficiently used to offload disproportionately impacted trusts, as it was theoretically possible to have 95.1% fewer trust-days at 100% mechanical ventilator bed capacity assuming load was better distributed. On the other hand, despite a reduction in overall capacity, G&A bed occupancy levels relatively infrequently reached 'unsafe' levels, even at the individual hospital level. This in part may explain why the field hospitals and independent sector care provider beds were never substantially used. Only a very small fraction of the theoretical maximum field hospital bed capacity was operationalised (1.23%). Similarly, despite signing a 14-week block contract with all of the major independent sector care providers valued at £235 million,[28] these beds too remained largely unoccupied, with less than 24% of the theoretical maximum beds days for established

| Table 2 | Field (Nightingale) hospital occupancy and capacity | | |
|---|---|---|---|
| **Nightingale hospital location** | **Occupied beds at peak (n)*** | **Maximum operational beds (n)*** | **Maximum theoretical capacity (beds)** |
| London (Excel Centre) | 66 | 112 | 4000 |
| Manchester (Convention Centre) | 47 | 72 | 1000 |
| Birmingham (National Exhibition Centre) | 0 | 0 | 2000 |
| Bristol (University of West England) | NA | NA | 1000 |
| Washington (Centre of Excellence for Sustainable Advanced Manufacturing) | NA | NA | 450 |
| Harrogate (Convention Centre) | 0 | 0 | 500 |
| Exeter (Westpoint Arena) | NA | NA | 200 |

*Several hospitals were formally opened, but never reported an occupied bed, as such they did not appear in the SitRep data set (denoted by NA in the table). Those that were in the data set but had no patients are denoted by '0'.
NA, not available; SitRep, situation report.

ventilators (ie, not including the 1012 theatre-specific mechanical ventilators) having been used.

## Context

Initial estimates suggested that an additional 30 000 mechanical ventilators would be necessary to accommodate the impact of the COVID-19 pandemic. These estimates were later updated to just 18 000 mechanical ventilators, from an estimated baseline of 8000 across the UK.[29] It is difficult to determine the accuracy of these projections, as they were made in the absence of the impact of non-pharmacological interventions. However, the results of our study suggest that, at the population level, UK-based models of ventilator and bed resource utilisation which integrated the impact of non-pharmacological interventions were actually remarkably accurate.[16 30] Arguably the most influential modelling study was that of the Imperial MRC (Medical Research Council) Centre for Global Infectious Disease Analysis group, where the authors clearly illustrate that with full 'lockdown' (ie, the suite of non-pharmacological interventions that were eventually instituted) critical care bed capacity would not be overwhelmed.[16] The nuance that this modelling study lacked was that it failed to explicitly incorporate the impact of unequal distribution of burden, which manifested in our data as specific hospitals and trusts reaching full occupancy, despite the fact that at the national level there were a substantial number of unoccupied beds.

This retrospective analysis also highlights some of the early incorrect assumptions made about the UK's baseline resource availability. For example, in contrast to ministerial statements suggesting that there were approximately 8000 ventilators in the UK prior to the pandemic,[29] our results identified only 4123 operational beds compatible with mechanical ventilation on the first day of reporting in England. Even after acknowledging that our value does not account for the devolved nations (Wales, Scotland and Northern Ireland), it is unlikely that the initial figures reported by members of the parliament truly reflected operational capacity, as that would suggest only 50% of such equipment was in England, despite it representing 84% of the UK population. Interestingly, the absolute increase in ventilator numbers due to government incentives (eg, the UK's Industrial Ventilator Challenge) is much more similar to our reported results.

## Strengths and limitations

There are several strengths to this study. For example, the use of an administrative (ie, 'SitRep') data that are a statutory collection by NHS England, via a well-established reporting mechanism that has been exploited for research,[31] confers robustness to the data. One example of how this robustness manifested is, unlike other attempts to collect data at a national level to inform the COVID-19 response plan in the UK,[32] the degree of missingness in the data used in this study was minimal (see online supplemental material). Moreover,

in light of the unique access to the raw 'SitRep' data, we have been able to not only present our results at the trust level, to which previous endeavours have been limited,[33] but rather have been able to present information at a much more granular layer (ie, hospital/site level), thus providing a much richer understanding of resource utilisation that is less prone to the diluent effects of higher level geographies. Finally, a further strength of this study is the relative simplicity of the analysis; there are no complex statistical methods used as the descriptive summaries presented are sufficient to describe the experiences of nationalised (single-payer) health system in a high-income economy during the first wave of the COVID-19 pandemic.

Notably though, there are also several limitations to the data set and our analysis. Principally we have no information on individual clinician and patient behaviour that will have inevitably influence these occupancy rates and thus cannot comment on these factors. Second, there are limitations inherent to the SitRep data. In particular, data were not available during February and early March, during which some early 'bed mobilization' was likely carried out, and thus our observation period does not cover the entirety of the first wave (however, we believe it is unlikely that this undermines the major findings of this study). Moreover, changes introduced in 'SitRep' data collection halfway through the reporting period limited our ability to investigate critical care bed occupancy, which was the third bed-specific potential concern identified by forecasting experts. The hospital-level results should also be interpreted with caution as they are an incomplete representation of the core trust-level information and thus may not truly reflect the exact position of each organisation; for example, the trust corresponding to the single site that achieved 100% G&A occupancy was never itself at 100% total occupancy. On a related note, the core weakness of the 'SitRep' data is that data are presented as a daily snapshot (at 08:00) and therefore are unable to capture the nuances of the hospital throughput; in essence, both under-reporting and over-reporting of occupancy are possible using this method. As such, any marginal results where hospitals are only just over one of the 'safe occupancy-level' thresholds should be interpreted with caution as they could represent reporting artefacts. Moreover, the use of the occupancy thresholds reflects a limitation of our analysis, in that a proxy for adverse outcomes had to be used given that the necessary information was not readily available to directly explore the relationship between occupancy and patient-level outcome. Finally, the results of this study may not be generalisable to other countries given that it is specific to the UK National Health System infrastructure and reporting systems; for example, it is difficult to draw comparisons with other countries as UK-specific factors such as reporting definitions are likely to mediate the hypothesised occupancy–mortality risk relationship, which will inevitably limit the ecological validity of these results in other geographical settings.

### Implications for policymakers and clinicians

This study illustrates the potential for near real-time results reporting by which to determine the need for and the effectiveness of government policies introduced to address resource utilisation-specific issues as a consequence of the COVID-19 pandemic. For example, due to an unequal distribution of the resource utilisation burden across England, many trusts spent a significant period of time operating above 'safe-occupancy' thresholds, despite the fact that in the vast majority of circumstances there was relief capacity in geographically co-located trusts (ie, at the STP level). Out of the 81 trust-days spent at 100% saturation of their mechanical ventilation beds (which pertains to 23 trusts reaching this threshold), on all but 5 days there was spare capacity at the corresponding STP level, which would have resulted in only 4 trusts reaching 100% saturation at any point (online supplemental SFigure 11). This reflects a key operational issue for policymakers to address in preparing for a potential second wave, and would have been identifiable if the SitRep data had been used for now-casting. Moreover, other policies for which these results may be relevant include the creation of the Nightingale (field) hospitals and independent sector network partnership. Our results suggest that the early investment and the creation of an operational field hospital and independent sector network may yield more overtly positive results in the winter, when G&A occupancy levels regularly exceed 92%[34]; however, during the first wave of the pandemic they were underutilised.

### CONCLUSION

Using administrative data submitted by all secondary care organisations in England, we can conclude that at the national level there was an adequate supply of all bed types throughout the first wave of the COVID-19 pandemic. However, the burden of need was not equally distributed, and thus in many cases local demand exceeded the supply of beds, especially where it concerned mechanical ventilation. Although several of the policies introduced by the government, both historical (ie, STPs) and pandemic-specific (eg, the independent sector block contract), could have potentially addressed this issue, there is evidence that these interventions were not optimally used. As such, we hope that this paper acts as exemplar for how routinely collected administrative data can be used to evaluate policy interventions, especially in the context of the COVID-19 pandemic, as well as highlighting the need for locally relevant (in lieu of national or regional summaries), near-real-time information on service use for operational decision making.

**Author affiliations**
[1]Warwick Medical School, University of Warwick, Coventry, UK
[2]The Alan Turing Institute, London, UK
[3]Institute of Health Informatics, University College London, London, UK
[4]Department of Statistics, University of Warwick, Coventry, UK
[5]The Institute of Biomedical & Clinical Science, University of Exeter, Exeter, UK
[6]Department of Statistics, Imperial College London, London, UK
[7]Diabetes and Endocrinology, Royal Devon and Exeter NHS Foundation Trust, Exeter, UK
[8]Health Data Research UK, London, UK
[9]The Zeeman Institute for Systems Biology & Infectious Disease Epidemiology Research, University of Warwick, Coventry, UK

**Acknowledgements** We thank NHS Improvement and NHS England for providing access to the SitRep data. We also thank Dr Bu'Hussain Hayee for sense-checking the final draft.

**Contributors** Based on the CRediT taxonomy, the authors of this study made contributions to this manuscript in the following ways: conceptualisation (BAM and SV); data curation (HW and SV); methodology (HW, BAM and SV); formal analysis (HW, BAM, JMD, SD and SV); project administration and supervision (BAM, MK and SV); visualisation (HW, BAM, JMD, NT, AM, AD and SV); resources (SV and MK); established data access (MK); writing the original draft (BAM and HW); and reviewing and editing the draft (all authors). The corresponding author and the senior author (SV) had full access to all data and attest to the integrity of the analysis. The decision to submit for publication was agreed by all authors. BAM and SV act as guarantors of the work as presented.

**Funding** The study was funded by UK Research and Innovation. BAM, SV and SD are supported by The Alan Turing Institute (EPSRC grant EP/N510129). JMD is supported by an Independent Fellowship funded by Research England's Expanding Excellence in England (E3) fund. SV is supported by the University of Warwick IAA funding. HW is supported by the Feuer International Scholarship in Artificial Intelligence. JMD, NT and AM are supported by the NIHR Exeter Clinical Research Facility.

**Map disclaimer** The depiction of boundaries on this map does not imply the expression of any opinion whatsoever on the part of BMJ (or any member of its group) concerning the legal status of any country, territory, jurisdiction or area or of its authorities. This map is provided without any warranty of any kind, either express or implied.

**Competing interests** AM declares previous research funding from Eli Lilly and Company, Pfizer and AstraZeneca. SV declares funding from IQVIA. All other authors declare no competing interests.

**Patient consent for publication** Not required.

**Ethics approval** Data used in this study were made available through an agreement between the University of Warwick and the Scientific Pandemic Influenza Group on Modelling (SPI-M), which were acting on behalf of the British Government. The study was reviewed and approved by the Warwick BSREC (BSREC 120/19-20).

**Provenance and peer review** Not commissioned; externally peer reviewed.

**Data availability statement** Data may be obtained from a third party and are not publicly available. Trust-level data will eventually be published by NHS England as a freely accessible data resource, but outputs have been delayed by the COVID-19 pandemic. For expedited or more granular access, requests need to be made directly to NHS England (contact via england.dailysitrep@nhs.net). All codes for this study are available on request.

**ORCID iDs**
Bilal Akhter Mateen http://orcid.org/0000-0003-4423-6472

Harrison Wilde http://orcid.org/0000-0002-4391-8204
John M Dennis http://orcid.org/0000-0002-7171-732X
Spiros Denaxas http://orcid.org/0000-0001-9612-7791

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
