## [Reviewer comments · BMJ Open]

ARTICLE DETAILS

TITLE (PROVISIONAL)	A descriptive analysis of hospital bed capacity and usage across secondary healthcare providers in England during the first wave of the COVID-19 Pandemic
AUTHORS	Mateen, Bilal Akhter; Wilde, Harrison; Dennis, John; Duncan, Andrew; Thomas, Nick; McGovern, Andrew; Denaxas, S; Keeling, Matt; Vollmer, Sebastian

VERSION 1 – REVIEW

REVIEWER	POON, Chin Man The Chinese University of Hong Kong Hong Kong, China
REVIEW RETURNED	09-Aug-2020

GENERAL COMMENTS	In this paper, Dr. Mateen and colleagues report detailed descriptive statistics about hospital bed capacity and usage across secondary healthcare providers during the first wave of COVID-19 epidemic in England by using secondary data in daily situation reports. I do appreciate the effort made by the authors in compiling the data and ensuring the data quality. However, as a descriptive survey lacking hypotheses to be tested, the authors may require harder effort in presenting the background and rationale for conducting this study, as well as the interpretation and discussion of the study results in their context. Such effort is crucial to make the paper readable and citable to readers outside England. Major comments: The spectrum of COVID-19 can range from asymptomatic infection to severe pneumonia with acute respiratory distress syndrome and death. In some countries, some COVID-19 cases might be managed in ambulatory or community settings. Do the authors have any clue on the proportion of confirmed COVID-19 cases being admitted to non-specialist secondary care providers in England? Were there any admission criteria? If yes, did the criteria change over time during the study period? Were all confirmed cases able to be admitted to the hospitals on the same day of COVID-19 diagnosis? All these factors would play a role on the bed occupancy in the study settings and affect the interpretation your results and conclusion. A bit more background information for these questions are required. The baseline bed occupancy is derived based on the mean availability between January and March 2020, covering the period when the winter influenza season in the United Kingdom usually takes place. A more appropriate choice of the baseline period would be the same calendar month as those in your study period in the previous year, if the data were available. Otherwise, you may want to provide additional explanation for your choice of the baseline
---

	period, as well as the interpretation of the study results. I understand that the study period covered by this paper is limited by the availability of the daily situation reports. It appears that the study period does not cover the entire first wave of COVID-19 epidemic in England. Please elaborate what period / phase of the first wave is covered in this paper, and how it would affect the interpretation of the study results. Otherwise, it may be hard to achieve the purpose that the authors suggest about understanding regional differences in hospital capacity for informing the responses to a second wave (Page 5 Line 58 - Page 6 Line 3). Minor comments: The journal generally recommends a maximum of five figures and tables, though this is flexible. There are now 9 figures in the paper, impacting upon the paper's "reading". I may suggest putting some of them (such as Figure 4 & 5, and those without much elaboration in your main text) into the Supplementary Materials for easier reading. Perhaps to include "England" in the title to specify the settings of this study. Page 3 Line 12: In the objective of the abstract, it might be misleading to quote the period starting from 31st January 2020, while data for this study was available from 27th March 2020. The authors present the results of unequal distribution of bed utilization. What is the possible reasons for such observations? I understand the reasons cannot be easily illustrated in such a descriptive survey. Is there any clue from field experience, expert opinion or literature? Is the inefficient networks for off-loading the disproportionately impacts trusts the only reasons (Page 14 Line 19)? To what extent the unequal distribution of bed utilization was influenced by varied COVID-19 caseload or epidemic in different regions? You may want to enrich the discussion part by addressing the possible reasons. Perhaps you may want to present the epidemic curve of confirmed COVID-19 cases in some of the graph, such that the readers can understand the development of the epidemic, in conjunction with the bed occupancy. Page 27: It may be more appropriate to start the y-axis from 0. Page 31: Is the surge of G&A beds towards the end of the study period relevant to COVID-19 epidemic in England? Any point for discussion based on the graph to match with your study objective? Page 33: The numbering for the figure is incorrect. Page 34: The numbering for the figure is incorrect.
--	---

REVIEWER	Boilève Institut Gustave Roussy, France
REVIEW RETURNED	25-Aug-2020

GENERAL COMMENTS	I read with great interest this study from Mateen and colleagues called "Hospital bed capacity and usage across secondary healthcare providers during the first wave of the COVID-19 Pandemic". Overall, this study is well conducted and very interesting, especially considering inadequate local supply of beds. Even if this is specific to the UK National Health System infrastructure, it is interesting to
--

	analyze how the different countries cope with COVID-19 first-wave pandemics. Please consider the following comments. Minor comments  1. Please define STP in the abstract and in the summary since this is not an obvious abbreviation. Please also add the explanation given in the results (aggregates of trusts) at first occurrence. 2. Please specify the number of ventilated beds occupied by COVID-19 patients and those by non-COVID-19 patients in the text of the results. 3. Were there any transfers of patients across UK to adapt to bed occupation ? (as it could be observed in France with transfers of patients from the East of France and from Paris region to less impacted regions) 4. A figure with the number of COVID-19 patients diagnosed, hospitalized and in ICU in UK within the same period of the presented data would be interesting.
--	---

VERSION 1 – AUTHOR RESPONSE

Response to Reviewer 1

First, and foremost, we'd like to thank the reviewer for taking the time to review our manuscript. Please find below an itemised set of responses to his comments, which we hope he will agree have led to a much improved manuscript.

1) In this paper, Dr. Mateen and colleagues report detailed descriptive statistics about hospital bed capacity and usage across secondary healthcare providers during the first wave of COVID-19 epidemic in England by using secondary data in daily situation reports. I do appreciate the effort made by the authors in compiling the data and ensuring the data quality. However, as a descriptive survey lacking hypotheses to be tested, the authors may require harder effort in presenting the background and rationale for conducting this study, as well as the interpretation and discussion of the study results in their context. Such effort is crucial to make the paper readable and citable to readers outside England.

>> We appreciate the reviewer's perspective on this. We would like to re-iterate that our stated hypothesis is provided at the end of the introduction (reproduced below for ease of access), and it is unclear to us whether the reviewer finds this inadequate in general, or not of sufficient interest to non-UK based individuals.

"Understanding regional differences in hospital capacity is fundamental to informing the UK's response to a second wave, as well as for elucidating how to safely wind down repurposed surge capacity such as operating theatres to allow other much needed clinical activity to restart.[22] However, other than a few isolated news reports of hospitals exceeding their ventilator capacity,[23] it is unclear how well the NHS as a whole managed to respond to the additional demand for beds over recent months. In this study, we sought to describe the pattern of bed occupancy in hospitals across England during the first wave of the COVID-19 pandemic."

>> If it is the latter, we in fact spent a non-trivial amount of time discussing how to present our motivation for doing this work, and concluded that: 1) this work is necessary for the UK to better understand what we can do differently, and thus we think the real-world local policy impact justifies putting the publication through the rigorous process of a peer-review; 2) we wanted to draw out lessons that would be robust to generalisation across geographical settings, and the only abstract result we felt met this threshold was the idea of regional and national summaries of bed availability of beds being insufficient to understand the heterogeneous impact of the pandemic. Given that it is entirely possible the latter point is not made clearly enough, we have added the following text to the end of the conclusion to make clear our intentions to readers further afield than the UK.

Original Text

“Using administrative data submitted by all secondary care organizations in England, we can conclude that at the national level there was an adequate supply of all bed-types throughout the first wave of the COVID-19 pandemic. However, the burden of need was not equally distributed, and thus in many cases local demand exceeded the supply of beds, especially where it concerned mechanical ventilation. Although several of the policies introduced by the government, both historical (i.e. STPs) and pandemic-specific (e.g. the independent sector block contract), could have potentially addressed this issue, there is evidence that these interventions were not optimally utilized.”

Revised Text

“Using administrative data submitted by all secondary care organizations in England, we can conclude that at the national level there was an adequate supply of all bed-types throughout the first wave of the COVID-19 pandemic. However, the burden of need was not equally distributed, and thus in many cases local demand exceeded the supply of beds, especially where it concerned mechanical ventilation. Although several of the policies introduced by the government, both historical (i.e. STPs) and pandemic-specific (e.g. the independent sector block contract), could have potentially addressed this issue, there is evidence that these interventions were not optimally utilized. As such, we hope that this paper acts as exemplar for how routinely collected administrative data can be used to evaluate policy interventions, especially in the context of the COVID-19 pandemic, as well as highlighting the need for locally-relevant (in lieu of national or regional summaries), near-real-time information on service use for operational decision making.”

2) The spectrum of COVID-19 can range from asymptomatic infection to severe pneumonia with acute respiratory distress syndrome and death. In some countries, some COVID-19 cases might be managed in ambulatory or community settings. All these [below] factors would play a role on the bed occupancy in the study settings and affect the interpretation your results and conclusion. A bit more background information for these questions are required.

- a) Do the authors have any clue on the proportion of confirmed COVID-19 cases being admitted to non-specialist secondary care providers in England?

>> We would like to re-assure the reviewer that this sample is representative of the vast majority of all bed types across England. Moreover, given that the alternative hospitals were typically either specialist orthopaedic, cancer, and Heart/lung units, as well as dedicated women's and children's hospitals, the proportion of COVID admissions they had were substantially lower. To ensure that this is reflected in the text, we have now included the below text in the supplementary methods section.

“Representativeness of Sample

Using the 2nd of May (randomly chosen) as an exemplar date, the non-specialist acute trusts to which we have restricted this survey represented 6,359 of the 6,866 beds (i.e. 92.6%) compatible with mechanical ventilation across England (comprising all institutions reporting to SitRep). Similarly, for all bed types, our sample represents 92.4% (i.e. 98,882 of the total 106,981 across England).”

- b) Were there any admission criteria? If yes, did the criteria change over time during the study period?

>> There was no nationally mandated admission criteria in the UK; that decision was left to the discretion of the emergency department physician or the clinician to whom the patient presented. This undoubtedly led to heterogeneity in practice, and we have thus included the below text into the limitations in the discussion to reflect this:

“Principally we have no information on individual clinician and patient behaviour that will have inevitably influence these occupancy rates, and thus cannot comment on these factors.”

- c) Were all confirmed cases able to be admitted to the hospitals on the same day of COVID-19 diagnosis?

>> This data was not available in this administrative dataset. There are national level figures on the turn-around times, but these have their own limitations and thus we see little value in including them in our report (especially because the publically available results tend to be cross-sections rather than a continuous audit). To reflect this limitation, we have added the below text into the methods section.

Original Text

“Occupancy is calculated based on the status of each bed at 08:00 each day, and then later separated by the proportion that had a positive COVID-19 test.”

Revised Text

“Occupancy is calculated based on the status of each bed at 08:00 each day, and then later separated by the proportion that had a positive COVID-19 test; there was no available information on the temporal relationship between admission and a positive test, and thus this data reflects some combination of community-acquired and nosocomial COVID-19.”

2) The baseline bed occupancy is derived based on the mean availability between January and March 2020, covering the period when the winter influenza season in the United Kingdom usually takes place. A more appropriate choice of the baseline period would be the same calendar month as those in your study period in the previous year, if the data were available. Otherwise, you may want to provide additional explanation for your choice of the baseline period, as well as the interpretation of the study results.

>> We appreciate the standard option would have been to choose a similar period in a previous year, but in this context where decisions were made on the data available (i.e. based on January's SitRep), in combination with the fact that the UK has faced a steady decline in bed numbers over the last decade, we felt that it was most appropriate to use the immediately prior period as the comparator as that would have been the equivalent of doing nothing. Also the classic flu seasons does not necessarily see a huge mobilisation in bed numbers but rather sees a different pattern of occupancy, and as such, we are less concerned about this issue. We have added the following text to the supplementary methods to explain our rationale for the baseline we chose.

“The choice of the period prior to the first wave of the pandemic instead of the historical baseline from 12 months prior was informed by two important piece of information: 1) the UK has experienced a gradual downward trend in bed numbers [1], and thus to be able to use the comparable period from 2019 we would have required an adjustment for that trend to produce a realistic baseline (there was a chance that we would have hypothesized there being more beds than were created after the first few weeks of mobilization by over-estimating the baseline number without this correction); 2) we deemed that use of the exact number of beds available at the time of operational planning (i.e. in February/early march) had greater ecological validity, as this was about reflecting the change from what we know was available rather than an abstracted version of what might have existed relative to similar periods in previous years.”

3) I understand that the study period covered by this paper is limited by the availability of the daily situation reports. It appears that the study period does not cover the entire first wave of COVID-19 epidemic in England. Please elaborate what period / phase of the first wave is covered in this paper, and how it would affect the interpretation of the study results. Otherwise, it may be hard to achieve the purpose that the authors suggest about understanding regional differences in hospital capacity for informing the responses to a second wave (Page 5 Line 58 - Page 6 Line 3).

>> This is a very legitimate concern, and thus we have added the following text to the limitations to reflect this issue.

“Secondly, there are limitations inherent to the SitRep data. In particular, data were not available during February and early March during which some early ‘bed mobilization’ was likely carried out, and thus our observation period does not cover the entirety of the first wave (however, we believe it is unlikely that this undermines the major findings of this study).”

4) The journal generally recommends a maximum of five figures and tables, though this is flexible.

There are now 9 figures in the paper, impacting upon the paper's "reading". I may suggest putting some of them (such as Figure 4 & 5, and those without much elaboration in your main text) into the Supplementary Materials for easier reading.

>> As suggested by the reviewer, we have moved figures 4&5 into the supplementary material.

5) Perhaps to include "England" in the title to specify the settings of this study.

>> This was also suggested by the editors, and thus we are happy to oblige both the reviewer and the editors and have changed the title accordingly. The revised title is as below:

Revised Title:

A descriptive analysis of hospital bed capacity and usage across secondary healthcare providers in England during the first wave of the COVID-19 Pandemic

6) Page 3 Line 12: In the objective of the abstract, it might be misleading to quote the period starting from 31st January 2020, while data for this study was available from 27th March 2020.

>> As previously where we added text to the limitations to reflect our appreciation for the point being made by the reviewer, we have also amended the abstract as follows to prevent any misinterpretation of how much of the first wave this study covers.

Original Abstract

Objectives: In this study, we describe the pattern of bed occupancy across England during the first wave of the pandemic, January 31st to June 5th 2020.

Design: Descriptive survey

Setting: All non-specialist secondary care providers in England

Revised Abstract

Objectives: In this study, we describe the pattern of bed occupancy across England during the peak of first wave of the COVID-19 pandemic.

Design: Descriptive survey

Setting: All non-specialist secondary care providers in England, from March 27th to 5th June 2020)

7) The authors present the results of unequal distribution of bed utilization. What is the possible reasons for such observations? I understand the reasons cannot be easily illustrated in such a descriptive survey. Is there any clue from field experience, expert opinion or literature? Is the inefficient networks for off-loading the disproportionately impacts trusts the only reasons (Page 14 Line 19)? To what extent the unequal distribution of bed utilization was influenced by varied COVID-19 caseload or epidemic in different regions? You may want to enrich the discussion part by

addressing the possible reasons.

>> To clarify, the inefficient networks are a suggestion of a policy proposal that might have alleviated the impact of the unequal distribution which we fully acknowledge was due to heterogeneous prevalence of the condition across England. We could expand on the different types of heterogeneity such as geographic, demographic (i.e. older individuals were more likely to be admitted with COVID, and thus the demography differed so probably the occupancy did too), but we are not sure what this adds. We are comfortable with the descriptive remit of this study and are delighted that it is already sparking discussion as to what the underlying factors might be, but we feel speculation isn't helpful at this point, as it distracts from the central argument that most did not even appreciate that there would be this degree of heterogeneity.

8) Perhaps you may want to present the epidemic curve of confirmed COVID-19 cases in some of the graph, such that the readers can understand the development of the epidemic, in conjunction with the bed occupancy.

>> We have now added the epidemic curve in a separate graph, as overlaying it did not look particularly visually appealing. Please see the updated figure 1.

9) Page 27: It may be more appropriate to start the y-axis from 0.

>> We have generated these figures and noted that the only effect is to compress the curves. Given that the axis are consistent across the two elements, and the scale is not a purposeful manipulation to support an unsubstantiated claim, we feel that this potential change only serves to detract from the manuscript by making one of the figures harder to read. As such, we have left these figures as is. If the editors or reviewer feels strongly about this point, we will of course gladly provide the amended figures.

10) Page 31: Is the surge of G&A beds towards the end of the study period relevant to COVID-19 epidemic in England? Any point for discussion based on the graph to match with your study objective?

>> We believe that it presents a winding down of the additional mobilised beds. However, we admit that we do not have a clear interpretation as to its meaning at present, and thus would prefer to leave it to the readers' discretion as to speculate on what it might mean.

11) Page 33: The numbering for the figure is incorrect.

>> Thank you for spotting this. We have now amended the error.

12) Page 34: The numbering for the figure is incorrect.

>> Thank you for spotting this. We have now amended the error.

Response to Reviewer 2

Overall, this study is well conducted and very interesting, especially considering inadequate local supply of beds. Even if this is specific to the UK National Health System infrastructure, it is interesting to analyse how the different countries cope with COVID-19 first-wave pandemics.

>> Again, we are very grateful to the reviewer for the time spent reviewing this manuscript!

1) Please define STP in the abstract and in the summary since this is not an obvious abbreviation. Please also add the explanation given in the results (aggregates of trusts) at first occurrence.

>> Thank you for highlighting this omission. We have now amended the text and the abstract as suggested.

2. Please specify the number of ventilated beds occupied by COVID-19 patients and those by non-COVID-19 patients in the text of the results.

>> As requested by the reviewer, we have now presented a high level overview of these values in the text.

Original Text

“Ventilated beds occupancy never exceeded 62% of this capacity (Figure 1), [...]”

Amended Text

“Ventilated beds occupancy never exceeded 62% of this capacity at a national level (Figure 1), and the proportion of occupied beds which contained patients with COVID-19 fluctuated between 30.4% and 76.0% over the course of the first wave, [...]”

3. Were there any transfers of patients across UK to adapt to bed occupation? (as it could be observed in France with transfers of patients from the East of France and from Paris region to less impacted regions)

>> Unfortunately a limitation of the data is that we cannot track individual patients and thus we do not know if they were transferred, nor are transfer numbers recorded in the SitRep.

4. A figure with the number of COVID-19 patients diagnosed, hospitalized and in ICU in UK within the same period of the presented data would be interesting.

>> A similar suggestion was made by reviewer 1, and we are happy to oblige – please see the amended figure 1.

VERSION 2 – REVIEW

REVIEWER	Chin Man Poon The Chinese University of Hong Kong, Hong Kong
REVIEW RETURNED	26-Nov-2020
GENERAL COMMENTS	The manuscript has been much improved following this round of revision. I appreciate the effort from the authors, who have clarified the issues raised by the reviewers and provided supplementary

	information. The implications of the study results are now clearly elaborated, and both strengths and limitations of the study are addressed. Great work! Below are some suggestions for further touching up: Page 3 Line 13: Add “the” before “first” in “... during the peak of the first wave of the COVID-19 pandemic” Page 3 Line 19: Delete bracket in “... in England, from March 27th to 5th June 2020)” Page 9 Line 23” “Analyses”, not “Analysis” Page 11 Line 5-9: Does this sentence describe the situation for G&A beds? If yes, please specify. Page 24 Line 58 & 60: The referencing format (superscript) is different from those for Figure 3 and 4 above (number in brackets). Page 26: There are two red lines in the figure (Figure 1) on the top. Please specify the data for which each line represents in figure legend. Are they daily number of cases and moving average of cases? Page 38: Is the numbering of this table correct? Should it be “STable 3”?
--	---

REVIEWER	Alice Boilève France
REVIEW RETURNED	16-Nov-2020

GENERAL COMMENTS	Comments have been addressed, I recommend publication of the manuscript.
--

VERSION 2 – AUTHOR RESPONSE

Response to Reviewer 1

Comments to the Author

The manuscript has been much improved following this round of revision. I appreciate the effort from the authors, who have clarified the issues raised by the reviewers and provided supplementary information. The implications of the study results are now clearly elaborated, and both strengths and limitations of the study are addressed. Great work!

>> We are very grateful to the reviewer for their comments, and are glad to hear that they have found the revisions acceptable. An itemised list of amendments addressing the minor omissions and grammatical errors identified by the reviewer can be found below.

1) Page 3 Line 13: Add “the” before “first” in “... during the peak of the first wave of the COVID-19 pandemic”

>> Thank you for identifying this error. It has now been amended.

2) Page 3 Line 19: Delete bracket in “... in England, from March 27th to 5th June 2020)”

>> Thank you for identifying this error. It has now been amended.

3) Page 9 Line 23” “Analyses”, not “Analysis”

>> Thank you for identifying this error. It has now been amended.

4) Page 11 Line 5-9: Does this sentence describe the situation for G&A beds? If yes, please specify.

>> Thank you for spotting this omission. We have now identified it as such.

5) Page 24 Line 58 & 60: The referencing format (superscript) is different from those for Figure 3 and 4 above (number in brackets).

>> Thank you for identifying this error. It has now been amended.

6) Page 26: There are two red lines in the figure (Figure 1) on the top. Please specify the data for which each line represents in figure legend. Are they daily number of cases and moving average of cases?

>> We have now clarified what the two curves represent in the figure legend.

7) Page 38: Is the numbering of this table correct? Should it be “STable 3”?

>> Thank you for identifying this error. It has now been amended.

Response to Reviewer 2

Comments to the Author

Comments have been addressed, I recommend publication of the manuscript.

>> Again, we are very grateful to the reviewer for their original comments, and are glad that they found our proposed revision acceptable.